

# Radiative impacts of the Australian bushfires 2019-2020 - Part 1: Large-scale radiative forcing

Pasquale Sellitto[1,2], Redha Belhadji[1], Corinna Kloss[3], Bernard Legras[4]

[1] Univ. Paris Est Créteil and Université de Paris, CNRS, Laboratoire Interuniversitaire des Systèmes Atmosphériques, Institut Pierre Simon Laplace, Créteil, France
[2] Istituto Nazionale di Geofisica e Vulcanologia, Osservatorio Etneo, Catania, Italy
[3] Laboratoire de Physique de l'Environnement et de l'Espace, Orléans, France
[4] Laboratoire de Météorologie Dynamique, UMR CNRS 8539, École Normale Supérieure, PSL Research University, École Polytechnique, Sorbonne Universités, École des Ponts PARISTECH, Institut Pierre Simon Laplace, Paris, France

*Correspondence to*: Pasquale Sellitto (pasquale.sellitto@lisa.ipsl.fr)

**Abstract.** As a consequence of extreme heat and drought, record-breaking wildfires developed and ravaged south-eastern Australia during the fire season 2019-2020. The fire strength reached its paroxysmal phase at the turn of the year 2019-2020. During this phase, pyro-Cb developed and injected biomass burning aerosols and gases into the upper-troposphere--lower-stratosphere (UTLS). The UTLS aerosol layer was massively perturbed by these fires, with aerosol extinction increased by a factor 3 in the visible spectral range in the Southern Hemisphere, with respect to a background atmosphere, and stratospheric aerosol optical depth reaching values as large as 0.015 in February 2020. Using the best available description of this event by observations, we estimate the radiative forcing (RF) of such perturbations of the Southern-Hemispheric aerosol layer. We use offline radiative transfer modelling driven by observed information of the aerosol extinction perturbation and its spectral variability obtained from limb satellite measurements. Based on hypotheses on the absorptivity and the angular scattering properties of the aerosol layer, the regional (at three latitude bands in the Southern Hemisphere) clear-sky TOA (top-of-atmosphere) RF is found varying from small positive values to relatively large negative values (up to -2.0 W/m$^2$), and the regional clear-sky surface RF is found to be consistently negative and reaching large values (up to -4.5 W/m$^2$). We argue that clear-sky positive values are unlikely for this event, if the aging/mixing of the biomass burning plume is mirrored by the evolution of its optical properties. Our best estimate for the area-weighted global-equivalent clear-sky RF is -0.35±0.21 (TOA RF) and -0.94±0.26 W/m$^2$ (surface RF), thus the strongest documented for a fire event and of comparable magnitude with the strongest volcanic eruptions of the post-Pinatubo era. The surplus of RF at the surface, with respect to TOA, is due to absorption within the plume that has contributed to the generation of ascending smoke vortices in the stratosphere. Highly reflective underlying surfaces, like clouds, can nevertheless swap negative to positive TOA RF, with global average RF as high as +1.0 W/m$^2$ assuming highly absorbing particles.





## 1 Introduction


There is widespread consensus that anthropogenic climate change is increasing the frequency and severity of wildfires, due to higher temperatures, lower rainfall and lower surface moisture (Smith et al., 2020). A markedly sensitive area for human-induced increase of fire risk is Australia (Dowdy et al., 2019). As a consequence of extreme heat and drought, record-breaking bushfires developed in Australia during the 2019-2020 wildfire season, also called the *Black Summer*. For this

event, fires concentrated and gradually intensified in South New Wales region, in south-eastern Australia, starting from September 2019, and lasted until March 2020. Perturbations in tropospheric trace gas composition, e.g. strongly enhanced concentrations of fire tracers as carbon monoxide (CO), hydrogen cyanide (HCN) and volatile organic compounds (VOC), were detected downwind the fires during the whole fire season (Kloss et al., 2021a). The fire activity reached its paroxysmal phase at the end of December 2019, with the development of a series of strong pyro-convective cumulonimbus clouds

(pyroCB) that injected a significant amount of gaseous and particulate pollutants in the upper-troposphere—lower-stratosphere (UTLS). These plumes had significant impacts on the stratospheric composition, including stratospheric ozone (Yu et al., 2021) and the stratospheric aerosol layer at the hemispheric scale (Khaykin et al., 2020). A stratospheric injection of smoke aerosols ranging from $0.4 \pm 0.2$ Tg (Khaykin et al., 2020) to 2.1 Tg (Hirsch and Koren, 2021) was estimated. This injection produced the largest documented perturbation of the stratospheric aerosol layer, in terms of the SAOD

(stratospheric aerosol optical depth), ever documented for a wildfire event and comparable with the perturbation of moderate stratospheric volcanic eruptions (e.g. the eruption of Raikoke 2019, Kloss et al., 2021b). One additional very specific feature of this event was the generation of stable and self-maintained smoke-charged anticyclonic vortices (Khaykin et al., 2020, Kablich III et al., 2020). The most intense of these vortices rose, due to solar heating, well into the stratosphere, reaching 35 km altitude by March 2020. Similar vortices were retrospectively found also for the British Columbia (Canada) fires in 2017

(Lestrelin et al., 2021).

The hemispheric perturbation of the aerosol layer due to the Australian wildfires 2019-2020 have likely produced a significant reduction of the incoming solar radiation in the Southern Hemisphere, at both low and high latitudes (Khaykin et al., 2020, Heinold et al., 2021, Hirsch and Koren, 2021, Yu et al., 2021). Different estimates of the shortwave radiative forcing (RF) for this event have been proposed using pure observational (Hirsch and Koren, 2021), hybrid

observations/modelling (Khaykin et al., 2020) and pure modelling (Heinold et al., 2021, Yu et al., 2021) approaches. Both clear-sky (Khaykin et al., 2020, Hirsch and Koren, 2021, Yu et al., 2021) and full-sky (Heinold et al., 2021) RF estimates were provided. Top of atmosphere (TOA) RF estimates ranged between relatively large negative values (-1.0 ± 0.6 W/m$^2$, over cloud-free oceanic areas, Hirsch and Kohren, 2021; -0.31 ± 0.09 W/m$^2$ global-equivalent clear-sky mean radiative forcing for February 2020, Khaykin et al., 2020) to near-zero (-0.03 W/m$^2$ global-equivalent clear-sky effective radiative

forcing, Yu et al., 2021) and relatively large positive values (+0.50 W/m$^2$ Southern-Hemispheric full-sky instantaneous radiative forcing, Heinold et al., 2021). The different radiative forcing estimates do not agree in absolute values and in sign of the forcing, which is linked to either a cooling, if negative, or a heating, if positive, of the climate system. These





differences might partially be linked to different assumptions on the optical properties of smoke aerosols in these estimates, as well as the underlying clouds conditions for the different estimates. In any case, further clarifications and insight on these

inconsistencies are called for.

In this paper, starting from the hybrid observations/modelling methodology used in Khaykin et al., 2020 (hereafter referred to as K20), we provide new updated estimates of the shortwave clear-sky instantaneous RF of the smoke plume of Australian wildfires 2019-2020 and we investigate the impact of different assumptions on the optical properties of the plume's particles on these estimates. We show under which conditions the smoke plume from this fire event produces positive or negative RF

and we discuss implications on the mixing state of observed or modelled aerosol layers, as well as the impact of the presence of underlying clouds. We show that previous RF estimates mentioned above are all consistent, based on different assumptions of biomass burning aerosols optical properties. Section 2 describes the data and methods used in this work and introduces the basis of offline radiative transfer modelling. Section 3 presents the observed hemispheric perturbation of the UTLS aerosol layer, which is key input to our RF estimates. Section 4 presents and discusses our new RF estimates.

Conclusions are drawn in Sect. 5.





## 2 Data and methods

### 2.1 Offline radiative transfer modelling driven by observations

The overarching idea in our RF estimates is to use the best possible measured information on the aerosol layer perturbations due to a specific and isolated forcer - here the Australian fires smoke emissions - as already applied in the past to other

localised aerosol sources like volcanic eruptions (e.g. by Sellitto et al., 2016; Sellitto et al., 2020; Kloss et al., 2021b) and feed this information into detailed radiative transfer calculations by offline modelling. This is aimed at exploiting the precision and flexibility of offline radiative transfer calculations, while constraining the forcer's emissions using observations. This approach contrasts with the widespread used online RF modelling, which is based on the use of modelled forcer's description and a simplified radiative transfer modelling, that is then normally carried out gridpoint-by-gridpoint.

The scheme of our offline radiative transfer approach is outlined in Fig. 1.



**Figure 1: Scheme of the offline radiative transfer modelling used in this work**

The radiative impact of the Australian wildfire plume is estimated by means of the equinox-equivalent daily-average

shortwave surface and TOA direct instantaneous RF, using the UVSPEC (UltraViolet SPECtrum) radiative transfer model in its libRadtran (library for Radiative transfer) implementation (Emde et al., 2016). In our procedure, the radiative transfer equation is solved with the SDISORT (spherical DISORT) scheme (Dahlback and Stamnes, 1991). The RF estimates are integrated between 300 and 3000 nm, starting from a 0.1 nm spectral resolution. We use used Kurucz (1992) solar flux input. The extra-aerosol atmospheric state is set using the AFGL (Air Force Geophysics Laboratory) summer mid- or high-latitudes

climatological standards, depending on the latitude range (Anderson et al., 1986). Clear-sky conditions are principally considered in this work. The shortwave surface albedo is set to 0.07, a typical value for sea surface (Briegleb and Ramanathan, 1982) as most of the fire plume disperse over ocean. The procedure is similar to that used in K20 with some differences outlined in this Section. Inputs to the offline modelling, for the fire-perturbed simulations, are provided by monthly average OMPS-LP aerosols extinction coefficient profiles at 675 nm, for January to April 2020, thus extending by

two months the temporal interval of K20 estimates, carried out for January and February 2020. As in K20, different latitude bands are considered separately, 15° to 25°S, 25° to 60°S and 60° to 80°S; the latitude band 80° to 90°S is excluded due to





the limitations of the OMPS observations geometry. We assume that the Australian fires have no impact in the northern hemisphere. The spectral variability of the aerosol extinction is represented using the measured monthly mean Ångström exponent from SAGE III/ISS, for January to April 2020. The merged OMPS/SAGE dataset for the aerosol spectral

extinction is further discussed in Sects. 2.2-2.3. The interaction of the radiation fields with the pyrogenic aerosols depends also on other optical properties of these particles that are not directly accessible from observations. In particular, the absorption and scattering properties of the aerosol layer must be represented in the radiative transfer modelling. Thus, different hypotheses have been considered for these non-measured optical parameters of fire aerosols. The absorptivity of the layer is modelled by means of the single scattering albedo (SSA), which is varied, in our calculations, from 0.80 to 0.95 with

0.05 steps. The angular distribution of the scattering is modelled using Heyney-Greenstein phase functions with asymmetry parameters (g) of 0.50 and 0.70. These values have been considered as spectral-independent in the shortwave spectral range. The interval covered by both SSA and g is chosen to cover a reasonable variability for biomass burning aerosol and their subsequent atmospheric evolution (see, e.g., Ditas et al., 2018). This variability of SSA and g extends the choice made in K20 (SSA=0.85-0.95, g=0.7) and is necessary to cover a larger interval of possible configurations of the fire plume and

make more complete sensitivity analyses. Biomass burning aerosols are more absorbing (SSA as low as 0.80) and smaller in size (g as low as 0.50) in fresh plumes mostly composed of black carbon (BC) and can get progressively less absorbing and larger in size as BC ages and mixes with other emitted species, e.g. by condensation of pyrogenic organic compounds or mixing with pyrogenic secondary organic aerosols or sulphates. In this case, during the aerosol layer evolution and mixing, BC can evolve towards brown carbon aerosol layers (BrC), which are characterised by less absorbing (SSA as high as 0.95)

and larger (g as large as 0.70) aerosol particles. Representing the atmospheric evolution of biomass burning aerosol and their evolving mixing state is challenging. These processes are usually regarded as not completely well represented in online aerosol/climate models; biomass burning aerosol have been demonstrated to be generally represented as too absorptive in models (Brown et al., 2021).

The RF of Australian fires plumes is estimated by comparing the outputs of the radiative simulations with fire-perturbed

aerosol layer to the reference case of a clean background. As a clean background, we select the OMPS/SAGE spectral extinction observations for the year 2019, for the respective four months of January to April. It is important to stress that the first months of the year are usually associated with the fire season in Australia. The simulations in 2019 are more representative of a "normal fire season" than a totally fire-aerosol-free atmosphere. Thus, our RF estimates can be regarded as the specific RF of the exceptionally strong 2019-2020 fire season in Australia with respect to a "normal" fire season. For

the background simulations, background SSA and g in the UTLS are considered (SSA=0.99 and g=0.70). Both fire-perturbed and background aerosol extinction profiles are discussed more in detail in Sect. 3.

We estimate RF at the TOA and at surface, for different solar elevations in terms of the solar zenith angle (SZA). The daily-average shortwave TOA RF for the fire-perturbed aerosol layer is calculated as the SZA-averaged upward diffuse irradiance for the background simulation minus that with the Australian fire aerosol perturbation, integrated over the whole shortwave

spectral range. The shortwave surface RF is calculated as the SZA-average downward global (direct plus diffuse) irradiance





with aerosols minus the background, integrated over the whole spectral range. We estimate equinox-equivalent daily average RF by assuming that the duration of day and night is equal.

## 2.2 Aerosol extinction observations with the Ozone Mapper and Profiler Suite – Limb Profiler (OMPS-LP)

The Ozone Mapping and Profiler Suite (OMPS) has flown onboard the Suomi National Polar-orbiting Partnership (Suomi-NPP) satellite since January 2012. The OMPS suite carries a Limb Profiler (OMPS-LP), which observes vertical profiles of scattered solar radiation, in the 290-1000 nm spectral range, in a limb geometry, i.e. tangent to the Earth's atmosphere and without looking directly towards the sun. Three detectors observe the limb radiation at slightly different angles; for the present work, we retain only measurements obtained with the central detector which are of higher quality (Ghassan Taha, personal communication). The OMPS-LP sounder is mainly designed to provide ozone concentration and aerosol extinction profile observations from cloud top to 60 km (for ozone concentration profiles) and 40 km altitude (for aerosol concentration profiles).

Building upon the v1.0 and v1.5 versions, an up-to-date v2.0 version of the aerosol extinction profile inversion algorithm has been recently developed (Taha et al., 2020). With respect to previous datasets versions, OMPS-LP aerosol extinction profile scheme v2.0 exhibits significant retrieval improvements, especially in the Southern Hemisphere (SH), when compared with independent datasets (Taha et al., 2020). This is very relevant for the present study which focuses on a purely southern-hemispheric event. Thus, as main input data to our offline RF estimates, the OMPS-LP v2.0 aerosol extinction observations at 675 nm have been used. An older OMPS-LP data collection (OMPS-LP USask, Zawada et al., 2018) was used in K20.

## 2.3 Ångström exponent observations with the Stratospheric Aerosol and Gas Experiment III on the International Space Station (SAGE III/ISS)

The spectral variability of the aerosol extinction is a key parameter towards the characterization of the interaction of radiation field with a given aerosol layer, and thus must be modelled in our offline RF calculations. While OMPS-LP v2.0 provides multi-spectral aerosol extinction, it is not recommended to use the different wavelength bands together since the accuracy of each wavelength retrieval is not homogeneous and is affected by its weighting function, stray light contamination and other sources of bias (Taha et al., 2020). The spectral variability of the aerosol extinction can be empirically modelled using the Ångström law and its Ångström Exponent (AE) parameter. Thus, in this work we have used the AE estimated using multi-spectral observations of the Stratospheric Aerosol and Gas Experiment III instrument on the International Space Station (SAGE III/ISS). The SAGE III/ISS instrument, operational since February 2017, provides stratospheric aerosol extinction coefficient profiles using solar occultation observations, i.e. by looking towards the Sun, at nine individual spectral bands from 385 to 1550 nm. The solar occultation geometry of SAGE III/ISS provides a better signal to noise ratio than the limb scattered sunlight geometry of OMPS, even if with a critically smaller horizontal sampling. Then, SAGE is not well adapted to monitor rapid spatial and temporal variations of the aerosol extinctions produced by isolated forcings, while it is an optimal source for the average information of its spectral variability. The AE used in this work is





derived using combinations of SAGE observations of the aerosol extinction at 521 and 869 nm. In K20, January 2020 average values for the January/February 2020 fire-perturbed runs, and prescribed AE values for the background, were used.

In the present work, we use respective SAGE III/ISS means for January to April 2020 and 2019, for fire-perturbed and background runs. The fire plume and background descriptions with OMPS/SAGE observations are then much improved in the present work with respect to K20.

## 3 Hemispheric perturbation of the stratospheric aerosol layer

Time series of the zonal average vertical distribution of the OMPS-LP aerosol extinction at 675 nm, in the latitude bands 15-

25°S, 25-60°S and 60-80°S, are shown in Fig. 2. With respect to what used for the RF estimates, these time series are extended to the whole year 2020, so to put our RF estimates for January-April within the larger temporal context. Due to the OMPS-LP geographical sampling and observations geometry, we exclude data from April to September 2020 in the latitude band 60-80°S. First enhancements of the UTLS aerosol extinction due to the Australian fires 2019-2020 are visible at the beginning of January in the band 15-25°S and, at lower altitudes, in the band 25-60°S. An initial injection of smoke aerosols

at top altitudes as high as 14 (Hirsch and Koren, 2021) and 17 km (K20) has been shown in previous works; Fig. 2 confirms this injection top altitude. The main aerosol perturbation subsequently extends at progressively increasingly higher altitudes, reaching about 20 km altitude, at 15-25°S and 25-60°S, and about 15 km, at 60-80°S. These differences in top altitudes of the plume may reflect differences in the tropopause height for these different latitude bands or be linked to sampling artifacts due to the average over a large latitude band. The perturbation is observed with a time lag of about 1 month at the

southernmost latitude band of 60-80°S due to the time needed for poleward transport associated with the Brewer-Dobson circulation (Butchart, 2014). The main perturbation is maximum in February 2020. Aerosol extinction values start to decrease from April 2020, and signatures of plume descent in altitude are visible for all latitude bands. Weaker ascending (January to April) and descending signatures (after April) are also observed at extremely high altitudes at all latitude bands, reaching altitudes as high as 35 km at 25-60°S. This is linked to the generation of ascending vortices in the stratosphere and

their descent, during this fire event, as discovered and described for the first time by K20, and will be further discussed in a companion paper.





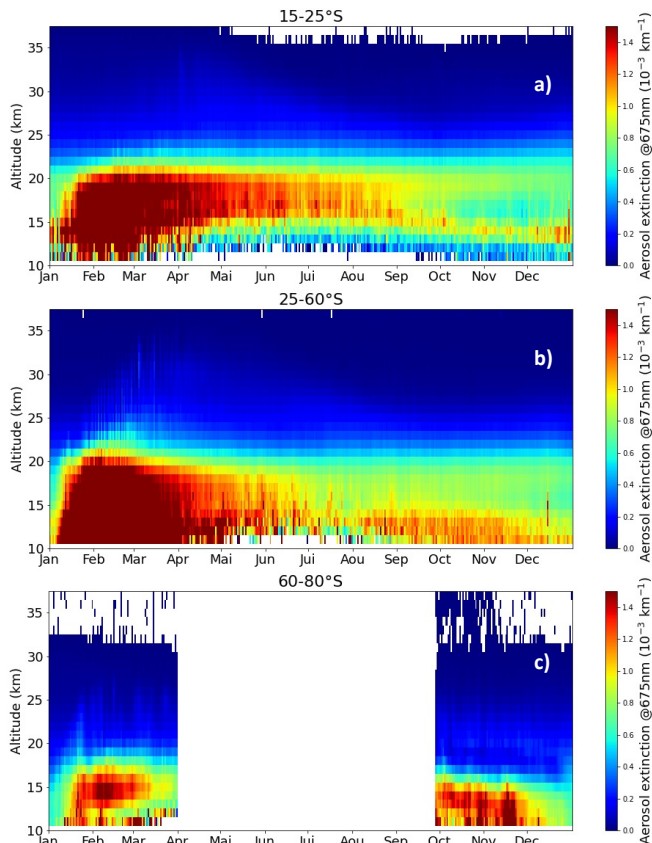

**Figure 2: Time series of the vertical profiles of the aerosol extinction at 675 nm from OMPS-LP observations. Time series are shown as zonal averages in the latitude interval 15-25°S (panel a), 25-60°S (panel b) and 60-80°S (panel c). The period between**
**April and September 2020 is excluded for the band 60-80°S due to limitations in the geographical sampling of OMPS-LP and its observation geometry.**

The time series between January and April 2020 of the zonal average integrated stratospheric aerosol optical depth (SAOD) at 675 nm, in the three latitude bands, is shown in Fig. 3. Time trends at all latitude bands point at a marked increase of the SAOD from the fire paroxysmal phase at the beginning of January to February 2020. The maximum of the SAOD is
observed in February 2020 at 25-60°S, where it reaches values as high as about 0.015. This value is comparable with peak values for SAOD perturbations associated with moderate stratospheric post-Pinatubo volcanic eruptions (e.g. Andersen et al., 2015, see its Figure 4). More limited February maxima are observed at 15-25°S (about 0.004) and 60-80°S (about 0.005). The SAOD values are then decreasing, more or less steeply depending on the latitude band, from February to April 2020. This trend, with a maximum about a month after the initial injection of the biomass burning aerosols, may be due to different
reasons: a) the OMPS detector gets saturated during the first phases of the event and then underestimates the aerosol extinction, b) the plume is progressively transported towards higher altitudes due to internal heating by absorption of solar radiation, c) the aging of biomass burning aerosol and the mixing with co-emitted aerosol, or secondary aerosols formed by co-emitted gaseous species, produces progressively more optically thick particles. This kind of temporal lag of the aerosol





extinction peak with respect to the occurrence of the paroxysmal event is often observed for stratospheric volcanic eruptions
(e.g. Haywood et al., 2010, Kloss et al., 2021b) and attributed to the build-up of the aerosol plume starting from the conversion of sulphur dioxide (SO2) emissions to secondary sulphate aerosols. For biomass burning plumes, the chemical composition of the gaseous and particulate emission can be significantly more complex than for volcanic plumes; the progressive formation of liquid secondary organic or sulphate aerosols can nevertheless bring to similar increases in the overall plume AOD by new particles formation and/or deposition/condensation over solid BC primary aerosol emission.
Thus, even if we cannot exclude any of the above hypotheses, we are inclined to consider the aging of the plume as an important factor at play towards the determination of the optical properties of the fire plume for this event.

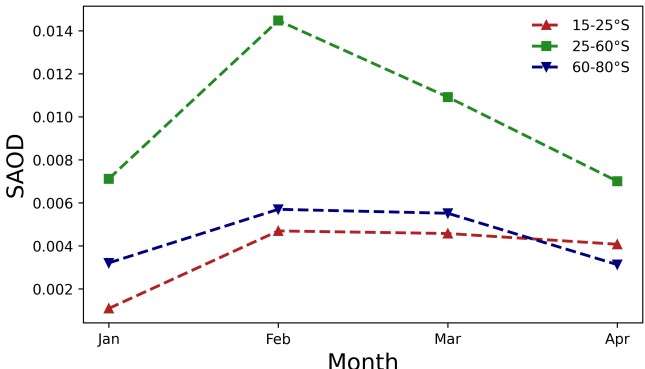

**Figure 3: Monthly mean OMPS-LP stratospheric aerosol optical depth (SAOD), from January to April 2020, in the latitude interval 15-25°S (red upwards triangles and line), 25-60°S (green squares and line) and 60-80°S (blue downwards triangles and line).**

The largest perturbation of the UTLS aerosol layer by Australian fires aerosol is observed, in OMPS data, for the month of February 2020 (see Figs. 2-3). Figure 4 shows the mean monthly February aerosol extinction profiles at 675 nm, for the fire-perturbed atmosphere (February 2020) and respective background atmosphere (February 2019), for the three latitude bands. The largest perturbation of the aerosol extinction, with respect to background, is observed in the UTLS of the 25-60°S
region, with extinctions reaching values as high as 0.004 km$^{-1}$ (background values in the range 0.0005-0.0008 km$^{-1}$). For the three latitude bands, the biomass burning aerosol injections due to the Australian fires 2019-2020 have produced an enhancement by a factor 2 to 3 in the UTLS. This enhancement extends up to over 20 km between 15 and 60°S, and is confined to altitudes lower than 20 km at 60-80°S. These mean monthly aerosol extinction profiles, for the months of January to April, are used in our RF calculations.





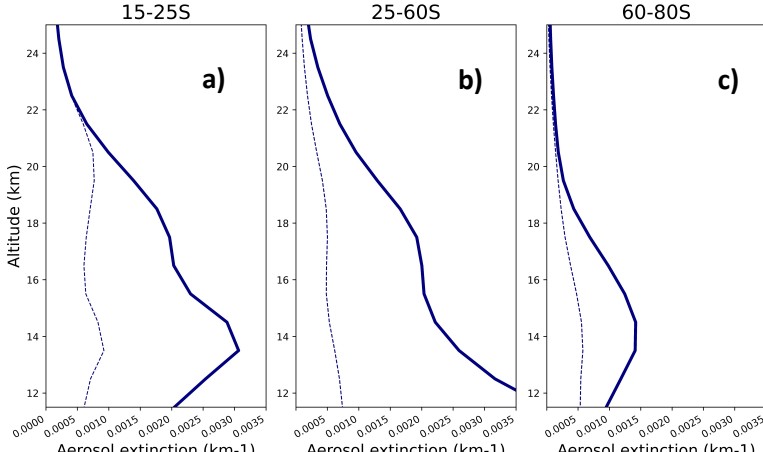

**Figure 4: Monthly mean OMPS-LP aerosol extinction coefficient profiles at 675 nm, for the months of February 2020 (perturbed profiles, solid lines) and February 2019 (background profiles, dashed lines), in the latitude interval 15-25°S (panel a), 25-60°S (panel b) and 60-80°S (panel c).**

## 4 Radiative forcing

### 4.1 Impact of the optical properties

An aerosol layer interacts with the radiation field by absorbing and scattering the radiation. These two interaction processes can be considered independent one from each other and add up to provide the extinction of the aerosol layer. Then, the ensemble optical properties of a given aerosol layer can be defined, in a compact way, by defining their overall extinction, absorption and scattering properties by means of the aerosol extinction (or optical depth), SSA (single scattering albedo) and scattering phase function. The SSA is defined as the ratio of the scattering and extinction cross sections of the layer and, as such, it quantifies the fraction of the total extinction of radiation by the layer that is due to scattering and absorption: smaller SSA values are linked to more absorbing particles. In the shortwave spectral region, the only strongly absorbing typology of aerosol is BC, with SSA reaching values as low as 0.80 or less. The scattering phase function defines the angular distribution of the scattered radiation by the aerosol layer. It can be compactly represented by the asymmetry parameter g which is the mean value of the cosine of the scattering angle, weighted using the phase function. As such, it is a compact indication of the direction of preferential scattering. With the assumption of spherical particles, for the Mie theory of the interaction of radiation with small particles, a more marked forward scattering, and then larger values of g, are typical of larger particles.

The TOA RF of a single aerosol layer depends on the properties of both the aerosol layer and the underlying surface. One raw parameterisation of the TOA RF is in Eq. 1 (Seinfeld and Pandis, 2016). Please note that in our offline modelling we have realised full radiative transfer simulations and Eq. 1 is only used here to discuss the general expected variability of the TOA RF as a function of the aerosol layer's optical properties. In Eq. 1, S is the solar irradiance input, T is the gaseous atmospheric transmissivity, SSA is the single scattering albedo, β is back-scattering fraction (larger β are associated with





smaller g) and $R_s$ is the surface reflectivity. While for purely non-absorbing aerosols like secondary sulphates (SSA approaching to 1.0) the TOA RF can theoretically only be negative, in case of absorbing aerosols (SSA lower than 1.0) the
TOA RF can be negative, zero or positive. The sign of the TOA RF arises from the competition of the two terms in parenthesis at the right hand of Eq. 1. Factors that facilitate the occurrence of positive (or less negative) TOA RF values are: the presence of strongly absorbing aerosol (small values of SSA), of large particles (small values of β and then large values of g) and reflective underlying surfaces (large values of $R_s$). In Eq. 1, multiple scattering is not accounted for.

$$\text{TOA RF} = -S\ T^2\ \text{SSA}\ \beta\ \text{AOD}\ \left((1 - R_s)^2 - 2R_s \frac{(1-SSA)}{\beta\ \text{SSA}}\right)$$ (1)

The surface RF accounts for both the variation of the radiation fields at TOA and the absorption and scattering processes within the Earth's atmosphere, due to the presence of the aerosol layer.

It is expected that a fresh fire plume, mostly formed of small-sized BC particles, has a small SSA (typical values about 0.80) and a small g (typical values about 0.5), in the shortwave spectral range. As the plume ages, the atmospheric evolution processes and mixing with other co-emitted species or subsequently formed secondary aerosols leads to less absorbing and
larger particles, thus increasing both SSA and g (e.g., Ditas et al., 2018, Brown et al., 2021). Figure 5 shows sensitivity analyses of the TOA and surface RF, for the four months of our full offline radiative simulations of the Australian fires, when using different assumptions on the SSA and g. Both TOA and surface RF depend critically on SSA and g assumptions in our offline calculations. Increasing SSA, i.e. more aged/mixed biomass burning plumes and then less absorbing aerosol layers, lead to stronger negative TOA RF and weaker negative surface RF. Increasing g, i.e. more aged/mixed biomass
burning plumes and then larger particles, on average, lead to less negative TOA and surface RF. Negative values of the RF as large as -2.0 (TOA RF) and -4.5 W/m² (surface RF) are found in more fire-affected regions (e.g. 25-60°S). While the surface RF is always negative in our experiments, positive TOA RF are in some cases obtained for extremely absorbing layers (SSA = 0.80) and/or large particles (g=0.7). This is particularly observed in January 2020. These results are consistent with the simple parametrisation of Eq. 1. It is well known that aerosol/climate models overestimate the absorptivity of
biomass burning plumes, due to an incomplete description of the plume aging and mixing state (Brown et al., 2021). Thus, the positive TOA RF values obtained for this event by Heinold et al. (2021), using an aerosol/climate model, can possibly be partially explained with this argument. On the contrary, our sensitivity analyses show that, in clear-sky conditions, for the observed aerosol extinction and its spectral variability, we obtain negative TOA and surface RF for almost all hypotheses on SSA and g – which are supposed to cover all possible conditions for these unmeasured optical properties. This might not be
the case in cloudy or full-sky (i.e. with partially cloudy sky) conditions. For high altitude plumes, of the type associated with this event, clouds are underneath the plumes and can thus be regarded as underlying surfaces with large $R_s$ in the shortwave range. From Eq. 1, this is a factor leading to less negative TOA RF, thus it is not excluded to have some positive RF in case of locally large cloud fractions. This is further discussed in Sect. 4.3.





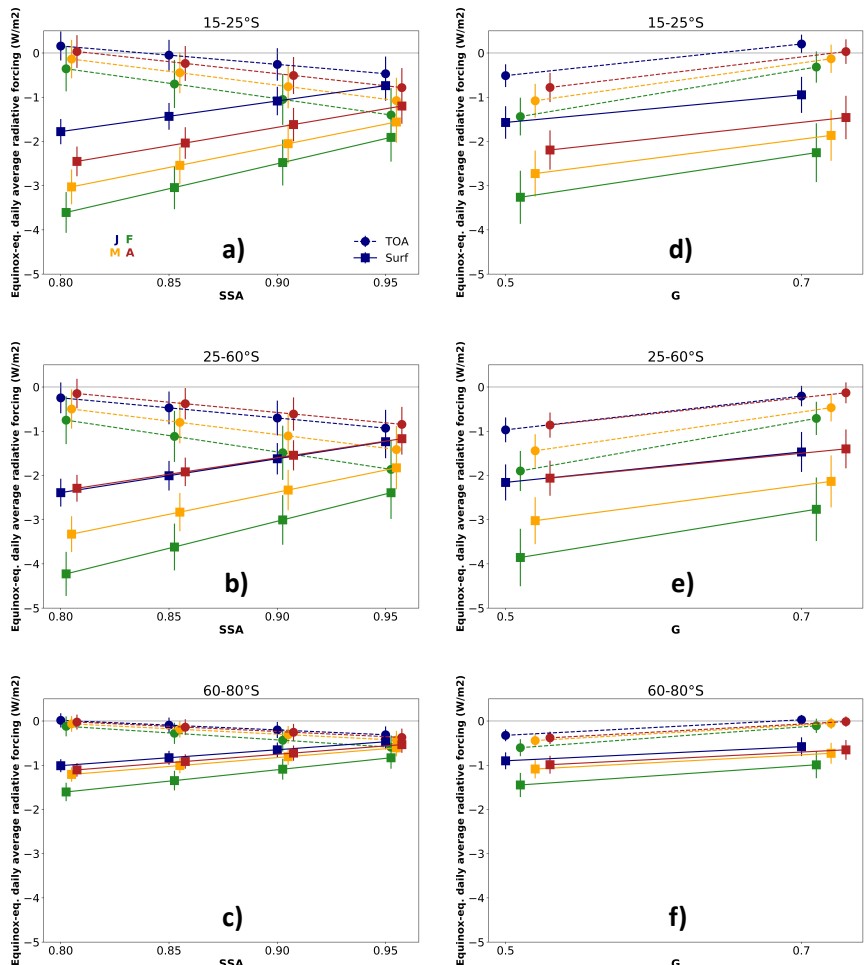

**Figure 5: Monthly mean regional equinox-equivalent daily average RF at TOA (dots and dashed lines) and surface (squares and solid lines) as a function of the SSA (panels a, b, c for 15-25°S, 25-60°S and 60-80°S, respectively) and g (panels d, e, f for 15-25°S, 25-60°S and 60-80°S, respectively). Different months are in different colours: January: blue, February: green, March: yellow, April: red.**

### 4.2 Global RF estimates

Starting from the regional estimates of the TOA and surface RF, global-equivalent RF can be estimated with an area-equivalent averaging of these values and the assumption that the Australian fires 2019-2020 did not impact the Northern Hemisphere. Figure 6 shows time series of these monthly mean (January to April 2020) area-weighted global-equivalent TOA and surface RF, as a function of the hypotheses on SSA and g, as well as an average of all scenarios. Table 1 also displays global RF estimates for different explicit scenarios of the bulk evolution of the biomass burning aerosol plume, including BC and BrC scenarios. From Fig. 6 it is possible to observe a clear trend of both TOA and surface RF, with a maximum RF in February 2020, then decreasing to approximately January values by April 2020. The surface TOA is





negative for each SSA/g hypothesis and each scenario. The TOA RF is negative, as well, for most hypotheses/scenarios and months, although it shows slightly positive values (maximum positive TOA RF of +0.07 W/m$^2$) in case of extreme values of SSA (0.80) and g (0.7), which means absorbing and large particles ("large BC" scenario in Tab. 1). We regard this scenario

as relatively unlikely, due to the aging processes of the biomass burning plume towards larger values of the SSA. Generally, BC particles are relatively small, then a value of g=0.5 would be more realistic; in this case, or even by considering a scenario of both large and small BC particles in the plume ("BC" scenario in Tab. 1), the TOA RF is negative during the whole four-months period, with a maximum of negative TOA RF of -0.18±0.16 W/m$^2$ in February 2020. This is also the scenario displaying the largest surface RF (-1.21±0.15 W/m$^2$ in February 2020). In the hypothesis of significant evolution of

the plume properties towards large and less absorbing particles (e.g. BrC scenario in Tab. 1), the TOA RF in February 2020 reaches negative values as low as -0.27±0.05 W/m$^2$. We also report on a scenario called "as K20" in Tab. 1. This scenario is obtained with the same assumptions of K20 (SSA=0.85-0.95 and g=0.7). Our new estimates, with the same conditions of K20, are revised at slightly smaller values than these previous estimates. For February 2020, we obtain -0.22±0.08 (TOA RF) and -0.69±0.15 W/m$^2$ (surface RF) (-0.31±0.09 and -0.98±0.17 W/m$^2$ in K20). This difference is due to the use of more

recent version of OMPS-LP aerosol extinction input, better estimate of the AE and of the background atmosphere in the present work. These scenarios exclude small and more reflective particles (g=0.5), that are considered in the average time series in Fig. 6 (red symbols and line) and the "All" scenario in Tab. 1. Due to the fact that aging/mixing of the plume is not explicitly represented in our work, and that SSA and g are not presently measured by regional/global satellite instruments so to be used as explicit inputs to our offline modelling, we regard this average, with its variability, as our best estimate of the

RF for this event. For this scenario (Tab. 1 but identical in the average case of Fig. 6) we have a maximum RF in February 2020 of -0.35±0.21 (TOA RF) and -0.94±0.26 W/m$^2$ (surface RF).

Our study confirms that the Australian fires 2019-2020 have produced a likely strong negative global RF at both surface and TOA, and that this effect has persisted for some months. The estimated RF is the strongest for documented fire events and comparable to the largest stratospheric volcanic eruptions in the post-Pinatubo era. The integrated long-term (dispersed)

TOA RF of the series of moderate post-Pinatubo volcanic eruptions has been estimated at $-0.19 \pm 0.09$ W/m$^2$ (Ridley et al., 2014) or smaller values (Schmidt et al., 2018). Individual recent stratospheric eruptions have been associated with peak global TOA RF of the order of -0.3 to -0.4 W/m$^2$ (Andersson et al., 2015), while the immediate post-eruption (so in the situation of still dense volcanic plume) TOA RF of the strong Raikoke eruption in 2019 (not covered by Andersson et al., 2015) has been estimated at values as large as $-0.38 \pm 0.06$ W/m$^2$ (Kloss et al., 2019). While the TOA RF impact of

Australian fires 2019-2020 is comparable to strong volcanic events, like the Raikoke eruption, the presence of carbonaceous partly-absorbing aerosols in fire plumes (which is not the case for the non-absorbing secondary sulphate aerosols that dominate volcanic plumes) has a yet more substantial impact on the surface RF, due to the large amount of shortwave radiation absorbed in the smoke plume.





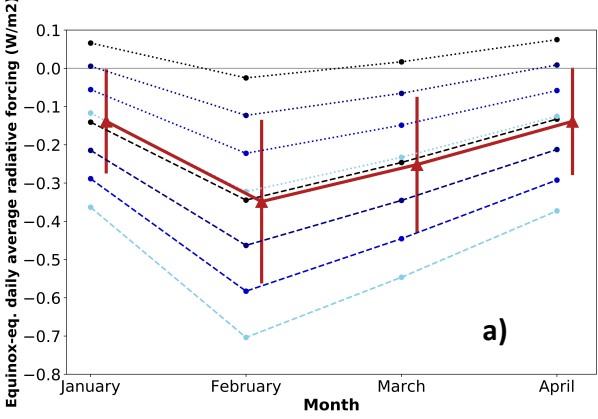

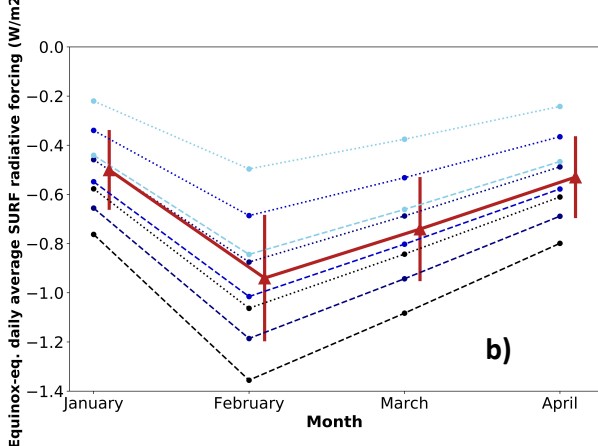

**Figure 6: Time series, from January to April 2020, of the monthly means area-weighted global-equivalent TOA (panel a) and surface RF (panel b) (based on equinox-equivalent daily average RF), for the UTLS perturbation of Australian fires 2019-2020. Different shades of blue dots and lines are for different SSA values, darker the shade smaller the SSA. Dashed and dotted blue lines are for g values of 0.5 and 0.7, respectively. The red triangles and thick solid lines represent the mean TOA and surface global-equivalent RF, averaged over the whole SSA and g scenarios. Error bars represent their variability.**





**Table 1: Monthly mean area-weighted global-equivalent TOA and surface RF (in W/m², for January to April 2020, for different scenarios of the unmeasured optical properties of the Australian fires plume. As K20: same assumptions of K20, SSA=0.85-0.95 and g=0.7. BC: black carbon, SSA=0.80 and g=0.5-0.7. Large BC: large black carbon, SSA=0.80 and g=0.7. BrC: brown carbon, SSA=0.90-0.95 and g=0.7. All: average of all scenarios, SSA=0.80-0.95 and g=0.5-0.7.**

| | January | February | Mars | April |
|---|---|---|---|---|
| **As K20** | | | | |
| TOA | -0.06±0.05 | -0.22±0.08 | -0.15±0.07 | -0.06±0.05 |
| SURF | -0.34±0.10 | -0.69±0.15 | -0.53±0.13 | -0.37±0.10 |
| **BC** | | | | |
| TOA | -0.04±0.10 | -0.18±0.16 | -0.11±0.13 | -0.03±0.10 |
| SURF | -0.67±0.09 | -1.21±0.15 | -0.96±0.12 | -0.70±0.09 |
| **Large BC** | | | | |
| TOA | +0.07 | -0.02 | +0.02 | +0.07 |
| SURF | -0.58 | -1.06 | -0.84 | -0.61 |
| **BrC** | | | | |
| TOA | -0.09±0.03 | -0.27±0.05 | -0.19±0/04 | -0.09±0.03 |
| SURF | -0.28±0.06 | -0.59±0.09 | -0.45±0.08 | -0.30±0.06 |
| **All** | | | | |
| TOA | -0.14±0.14 | -0.35±0.21 | -0.25±0.18 | -0.14±0.14 |
| SURF | -0.50±0.16 | -0.94±0.26 | -0.74±0.21 | -0.53±0.17 |

**4.3 On the impact of clouds**

All simulations discussed in Sect. 4.2 have been performed for clear-sky conditions and with a typical value of the surface reflectance $R_s$ for ocean surface ($R_s$=0.07, constant in the shortwave range). The plume disperses over the Southern Hemisphere, which is predominantly characterised by a predominant sea surface and a limited land surface cover. Following Eq. 1, a reflective underlying surface such as an opaque cloud can facilitate the occurrence of smaller negative or positive

TOA RF. To explore the role of clouds in the RF for this event, as a first approximation we simulate cloudy conditions by realising RF simulations using a very reflective underlying surface. To test the impact of clouds we have thus run the scenarios BC and BrC (see Tab. 1 for their definitions in terms of SSA and g) with a $R_s$ = 0.5. Such a shortwave albedo is met for various cloud types, including stratus and altostratus (Houze, 1993). We assume that the biomass burning plume and





clouds do not interact, are located at vertically separated levels and that the biomass burning plume is at higher altitudes than

clouds. The TOA and surface RF for BC and BrC scenarios, in presence of clouds, for the four months of January to April 2020, are summarised in Tab. 2. The TOA RF can have large positive values, under these conditions, especially for very absorptive particles and the BC scenario. In this case, average TOA RF for February 2020 can reach values as high as +1.0 W/m$^2$. For aged, much less absorbing particles, switching to the BrC scenario, the positive TOA RF is more limited, reaching a maximum values of about +0.3 W/m$^2$ in February 2020. For both scenarios, the TOA RF stays positive for the

whole fire-perturbed period under investigation. Consistently negative values of the surface RF, even if smaller in magnitude with respect to clear-sky conditions, are found for both scenarios. It's important to notice that the presence of clouds would affect the surface RF in a more complex way than just assuming a highly reflective underlying surface, i.e. by the reduction of transmission of radiation due to the physical presence of a cloud layer, thus surface RF of Tab. 2 are to be taken with extreme caution. It is also important to notice that this study of the clouds effect on the Australian-fire-induced RF assumes a

homogenous and constant cloudy sky for the whole four months, which is not a realistic configuration for a proper full-sky conditions, where the average cloud cover should be taken into account as a function of time and latitude. Further studies are ongoing for the estimate of full-sky RF with offline radiative transfer modelling.

**Table 2: Monthly mean area-weighted global-equivalent TOA and surface RF (in W/m$^2$), for January to April 2020, for different scenarios of the unmeasured optical properties of the Australian fires plume (see definitions in Tab. 1), with underlying clouds modelled as surface with large $R_s$ reflectivity ($R_s$=0.5).**

|  | January | February | Mars | April |
|---|---|---|---|---|
| **BC with underlying clouds** | | | | |
| TOA | +0.61±0.04 | +0.96±0.05 | +0.80±0.04 | +0.65±0.03 |
| SURF | -0.63±0.05 | -1.07±0.06 | -0.87±0.06 | -0.66±0.05 |
| **BrC with underlying clouds** | | | | |
| TOA | +0.19±0.09 | +0.28±0.15 | +0.24±0.12 | +0.20±0.10 |
| SURF | -0.23±0.07 | -0.43±0.11 | -0.34±0.09 | -0.24±0.07 |

## 5 Conclusions

In this manuscript we have presented an array of coupled observations/modelling simulations of the radiative transfer through the biomass burning plume linked to the record-breaking Australian fires 2019-2020. Realistic description of the

plume is provided with limb satellite observations of the UTLS aerosol extinction perturbation produced during and after the paroxysmal phase of this event (generation of pyroCb clouds overshooting to the UTLS and the stratosphere), compared to a background atmosphere, for the time period from January to April 2020. Aerosol observations are used as inputs to a



detailed and flexible offline radiative transfer modelling, to produce regional and global clear-sky TOA and surface RF estimates. Different hypotheses on the plume evolution have been considered, mirrored by the evolving unmeasured optical

properties of the plume, namely the SSA/absorptivity of the plume and the g/angular distribution of the scattered radiation. Aerosol extinction observations hint at the possible evolution of the plume, with an increasing extinction after the UTLS injection, pointing at aging and possible mixing of the emitted BC with other aerosols typologies, e.g. secondarily formed organics or sulphates from pyrogenic emissions of their precursors. We obtain TOA and surface RF that depend critically on the hypotheses on the optical properties of the plume and, so, on its atmospheric evolution. Depending on SSA/g

assumptions, we obtain different clear-sky RF estimates, with a general indication that the clear-sky TOA and surface RF are negative, thus leading to a cooling of the climate system. The only case where we obtain a (small) positive TOA RF is for extremely absorbing and large BC particles, which is a scenario that we estimate as relatively unlikely. In case of cloudy sky, modelled as an underlying highly reflective surface, both BC and more aged and reflective BrC scenarios are characterised by relatively large and positive TOA RF. These results possibly explain why recent papers on this event obtained positive

TOA RF using online aerosol/climate models. Aerosol/climate models are well known to overestimate the absorptivity of biomass burning aerosol through an incomplete description of their aging/mixing processes. In addition, full-sky RF estimates can lead, for large average cloud cover and absorbing particles, to even more positive TOA RF. Our best estimate of the instantaneous area-weighted global-equivalent radiative impact of the Australian fires 2019-2020 plume, in clear-sky conditions, is -0.35±0.21 (TOA RF) and -0.94±0.26 W/m$^2$ (surface RF). This confirms previous studies that rank this event

as the strongest radiative balance perturbation at TOA documented for a fire event and comparable with the strongest volcanic eruptions of the post-Pinatubo era. The impact at surface is yet stronger due to the additional absorption within the plume itself, which has generated rising smoke vortices into the stratosphere and will be investigated further in a companion paper.

**Data availability**

OMPS-LP v2.0 and SAGE data are freely available via the NASA-Earthdata portal (https://search.earthdata.nasa.gov/search).

**Acknowledgments**

This research has been supported by the Agence Nationale de la Recherche (grant no **21-CE01-0007-01**, ASTuS). The providers of the libRadtran suite (http://www.libradtran.org/) are gratefully acknowledged. Ghassan Taha, NASA, is

gratefully acknowledged for the help with OMPS data.



**Authors contributions**

P.S. designed the study and ran the RF simulations. R.B. and C.K. helped with SAGE/OMPS input data. All authors participated to the discussion of the RF results. P.S. wrote the manuscript and all authors participated to its revision and the editing.

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
