# Peer review of "Radiative impacts of the Australian bushfires 2019-2020 - Part 1: Large-scale radiative forcing"

_EGUsphere, 2022_

## Referee Comment (RC1)

**Radiative impacts of the Australian bushfires 2019-2020 - Part 1: Large-scale radiative forcing**

**Authors:** Pasquale Sellitto, Redha Belhadji, Corinna Kloss, Bernard Legras

**EGUSPHERE-2022-42**

In this manuscript the authors provide results concerning TOA and surface radiative forcing over the Southern Hemisphere due the impact of the extreme bushfires occured during the fire season of 2019-2020 in Australia.

The authors presented a highly valuable data set based on two different instruments of global coverage, the aerosol extinction observations with the Ozone Mapper and Profiler Suite – Limb Profiler (OMPS-LP) and the Ångström exponent observations with the Stratospheric Aerosol and Gas Experiment III on board of the International Space Station (SAGE III/ISS), and the importance of using experimental data as input into radiative transfer models. The authors presents new results of the shortwave clear-sky instantaneous radiative forcing of the Australian fires smoke plume and investigate the impact of different assumptions on the aerosol extinction, single scattering albedo and the asymmetry parameter g of the fire plumes. The manuscript is well written and the results are very important considering the positive or negative radiative forcing impacts on the Southern Hemisphere due different assumptions on the aerosol optical properties and also due clear-sky and cloud conditions. I recommend the article for publication following correction and clarification of a few minor issues described below.

**Comments and suggestions**

In the lines 100 to 102 the authors claim that "*As in K20, different latitude bands are considered separately, 15° to 25°S, 25° to 60°S and 60° to 80°S; the latitude band 80° to 90°S is excluded due to the limitations of the OMPS observations geometry.*" Please, consider to provide a earth map figure in order to ilustrate the overall region considered in the RF calculation.

It is not clear if the impacts of RF is for the whole Southern hemisphere. Could the RF results be different over Atlantic region between African and Sputh America Continents?

In lines 115 to 118 the authors state that "*Biomass burning aerosols are more absorbing (SSA as low as 0.80) and smaller in size (g as low as 0.50) in fresh plumes mostly composed of black carbon (BC) and can get progressively less absorbing and larger in size as BC ages and mixes with other emitted species, e.g. by condensation of pyrogenic organic compounds or mixing with pyrogenic secondary organic aerosols or sulphates.*" Please, consider to include some references here.

When the authors presented the variables used in sections 2.2 and 2.3, i.e., the aerosol extinction from the OMPS-LP and the Ångström exponent from SAGE III/ISS, it was not clear if there is an overlap between the measurements of both satellites and if the absence of the overlap affects in some way the RF results.

In figure 5 of pag. 12, please, consider to include a colour label inside the graphics (January: blue, February: green, March: yellow, April: red).

In figure 6 of pag. 14, consider to include a colour label inside the graphics for different values of SSA (darker the shade smaller the SSA).

---

## Author Comment (AC1)

**Reply to the two anonymous Reviewers of the manuscript "Radiative impacts of the Australian bushfires 2019-2020 - Part 1: Large-scale radiative forcing".**

Dear Editor, dear Reviewers,

We would like to thank the two anonymous Reviewers for their kind words about our work and their constructive criticism. There is an exciting debate about the Australian fires 2019-2020, the plume's aerosols physicochemical and optical evolution and their impacts. We revised our manuscript based on Reviewers' comments and we provide a point-by-point reply in the following. Please note that some interconnected comments have been aggregated and we provide a unified reply to these comments. When needed, we refer to the revised text using line numbering (Lxxx) and to the specific comments of the two Referees with their progressive numbering (SCxx). We think that our manuscript has improved thanks to the modifications, clarifications and the addition of a number of recent references (in some cases published between our first submission and the present review round – which shows how vital is this debate).

Sincerely,
Pasquale Sellitto on behalf of all coauthors.

**Reviewer #1**

**General Comment:**
In this manuscript the authors provide results concerning TOA and surface radiative forcing over the Southern Hemisphere due the impact of the extreme bushfires occured during the fire season of 2019-2020 in Australia. The authors presented a highly valuable data set based on two different instruments of global coverage, the aerosol extinction observations with the Ozone Mapper and Profiler Suite – Limb Profiler (OMPS-LP) and the Ångström exponent observations with the Stratospheric Aerosol and Gas Experiment III on board of the International Space Station (SAGE III/ISS), and the importance of using experimental data as input into radiative transfer models. The authors presents new results of the shortwave clear-sky instantaneous radiative forcing of the Australian fires smoke plume and investigate the impact of different assumptions on the aerosol extinction, single scattering albedo and the asymmetry parameter g of the fire plumes. The manuscript is well written and the results are very important considering the positive or negative radiative forcing impacts on the Southern Hemisphere due different assumptions on the aerosol optical properties and also due clear-sky and cloud conditions. I recommend the article for publication following correction and clarification of a few minor issues described below.
We thank the Reviewer for the kind words and the appreciation to our work.

**Specific Comments:**
1) In the lines 100 to 102 the authors claim that "As in K20, different latitude bands are considered separately, 15° to 25°S, 25° to 60°S and 60° to 80°S; the latitude band 80° to 90°S is excluded due to the limitations of the OMPS observations geometry." Please, consider to provide a earth map figure in order to ilustrate the overall region considered in the RF calculation. It is not clear if the impacts of RF is for the whole Southern

hemisphere. Could the RF results be different over Atlantic region between African and Sputh America Continents?

As stated at L293-295, based on the three different regional RF estimations in the mentioned latitude bands, we estimate here the global-equivalent RF: "global-equivalent RF can be estimated with an area-equivalent averaging of these values and the assumption that the Australian fires 2019-2020 did not impact the Northern Hemisphere". This is a quite classic and straightforward RF metrics, so we don't think further explanation in the manuscript or even a map is necessary.

2)  In lines 115 to 118 the authors state that "Biomass burning aerosols are more absorbing (SSA as low as 0.80) and smaller in size (g as low as 0.50) in fresh plumes mostly composed of black carbon (BC) and can get progressively less absorbing and larger in size as BC ages and mixes with other emitted species, e.g. by condensation of pyrogenic organic compounds or mixing with pyrogenic secondary organic aerosols or sulphates." Please, consider to include some references here.

There is a relatively large literature on the biomass burning aerosol aging and progressive physicochemical transformation of black to brown carbon. For the sake of conciseness, we have decided to cite just one work which we have found very informative here (and elsewhere, see Reviewer #2's SC8-9), Konovalov et al. 2021, which is now cited at L119.

3)  When the authors presented the variables used in sections 2.2 and 2.3, i.e., the aerosol extinction from the OMPS-LP and the Ångström exponent from SAGE III/ISS, it was not clear if there is an overlap between the measurements of both satellites and if the absence of the overlap affects in some way the RF results.

As discussed in Sect. 2.1, in our RF simulations, we use monthly means OMPS aerosol extinction and SAGEIII Ångström exponent for three latitude bands (15-25°S, 25-60°S, 60-80°S). Thus, yes, there are overlaps in the sense of this spatiotemporal average.

4)  In figure 5 of pag. 12, please, consider to include a colour label inside the graphics (January: blue, February: green, March: yellow, April: red).

Done. Figure 5 has been modified according to this comment and Reviewer #2's SC11.

5)  In figure 6 of pag. 14, consider to include a colour label inside the graphics for different values of SSA (darker the shade smaller the SSA).

Done.

**Reviewer #2**

**General Comment**

Although there are already numerous publications on the radiative impact of Australian smoke on the market, this manuscript adds new aspects and discusses inconsistencies in previous articles and thus is worthwhile to be published in ACP.
I have only minor points.

We thank the Reviewer for the kind words and the appreciation to our work.

**Specific Comments:**

1) page2, line38: Please specify.... whole fire season.... was that from September 2019 to January 2020 or from July 2019 to March 2020?
We mean September 2019 to March 2020. This is now explicitly stated at L38.

2) p2, l42: Meanwhile there are many smoke-ozone papers in addition to Yu et al., 2021, who presented not more than a few hypothetic sentences. Now we have in addition: Solomon et al., PNAS, 2022, Bernath et al., Science, 2022, Rieger et al., GRL, 2022, Ohneiser et al., ACPD, 2022, Ansmann et al., ACPD 2022.
Yes, these were not available at the time of our manuscript submission, and we cite them now at L42.

3) p2, l42-32: I suggest to cite also : Peterson et al., 2021, Australia's Black Summer pyrocumulonimbus super outbreak reveals potential for increasingly extreme stratospheric smoke events, npj Clim Atmos Sci, 2021, doi=10.1038/s41612-021-00192-9 In this paper, a nice summary of this record-breaking Australian pyroCb event (29 December 2019 to 5 January 2020) is given including an estimate of 1.1 Tg of emitted smoke.
This is now cited at L44-45.

4) P4, l99: Can you provide numbers regarding the extinction coefficient measurement range that OMPS-LP at 675 nm can measure? Probably 20 Mm-1 along the line of sight is too high (saturation effect) and 0.1 Mm-1 may be too low (no longer distinguishable from clear air...)? But these are speculations, you should know the measurement range.... and thus should be able to provide numbers.

5) P6, section 2.2 Or provide the extinction measurement range here....
Even if we do not have a specific instrumental expertise on OMPS-LP, we think that the upper limit mentioned by the Reviewer #1 before reaching saturation ($20*10^{-3}$ km$^{-1}$) is a bit too high and in any case quite far from our measured range (Fig. 4, average values not exceeding $3*10^{-3}$ km$^{-1}$ in the plume vertical range).

6) p8, Figure 2: I have a few questions to this figure! 25-60S: You probably had saturation effects in the height range from 10-15 km in January and February 2020. On the other hand, are you able to detect cirrus at heights up to about 12 km?
We don't exclude the possibility to have saturation effects (this is discussed at L205-219), even if the observed average aerosol extinction profiles display values not exceeding ~$3*10^{-3}$ km$^{-1}$. Clouds are ideally pre-screened in OMPS-LP data (Taha et al., 2021) even if thin cirrus extinction might be present in some cases. Relatively large average Ångström exponent values in our SAGEIII/ISS dataset seem to suggest a limited cirrus clouds impact.

7) One should check the lidar long-term observations at Punta Arenas at 53S (Ohneiser et al., ACPD, 2022). Lidar does nor suffer from saturation effects. This lidar data set is for ONE single site, however, should be in general agreement with the development of the smoke extinction and AOD (as given here for latitudinal belts) in January to March 2020. The lidar extinction values are for 532 nm, and can be translated into 675 nm values by using the Angstroem exponents.

We fully agree that the long-term LiDAR observations at Punta Arenas are of a great importance in this context. Nevertheless, from our perspective, it is very difficult to quantitatively compare our large-scale average aerosol extinction observations with point observations at one specific site. In the paper by Ohneiser et al., it is clearly stated that the smoke conditions were very variable at this location. In addition, the wavelength conversion through the Ångström law is not trivial due to the variability of the Ångström exponent associated with the physicochemical evolution of the smoke plume. By the way, we will surely explore the possibilities of comparisons of the two datasets in the future.

8) 15-25S: The same here for the height range from 10-15 km in January and February 2020. Can we trust the January 2020 data?

9) p9, Figure 3 corroborates my 'opinion'. There is a quite nice decay behavior from February to April. And because the smoke injection was in the beginning of January (not in the middle or the end of January). Why are the January data NOT in line with the general trend from Februray to April?

The trend of Fig. 3 is discussed at L205-219. We very briefly summarise this discussion here. We propose for this trend three possible reasons: 1) saturation of the OMPS detector, 2) dynamical features, like the radiatively-driven lofting of the plume, 3) the plume's physicochemical evolution, due to the progressive formation of secondary aerosols and mixing with black carbon, leading to brown carbon aerosols. For this latter possibility, please note that an increase in AOD, and a simultaneous increase in SSA, are often observed during biomass burning plumes atmospheric ageing (a spectacular case is shown in Konovalov et al., 2021, briefly discussed now at L216-219). Thus, we don't see any reason to reject the trend shown in Fig. 3, or to attribute it to purely an instrumental artifact. It is important to notice the general philosophy of this manuscript and the fact that we organised the specific radiative forcing estimations as a series of sensitivity analyses, since none of the three possibilities discussed above can be excluded a priori (or can even be at play at the same time).

10) p10, Figure 4: Can you explain, why there is steady decrease of extinction coefficient from 12-24 km, but not in the belts 15-25S and 60-80S? Again, are you sure that all extinction is purely caused by smoke (no cirrus)? What about January profiles? Probably, signal saturation effects should show up in the extinction profiles. I am so critical or suspicious in these points.... because in section 4 the radiative forcing results are shown, and the main question arises: Do these well and carefully performed simulations reflect the REALITY or the shortcomings (especially wrong January extinction profiles) in the satellite observations.

The region initially affected by the plume injection is indeed 25-60°S (the pyroCb In January developed in an area between about 25 and 35°S). There is of course stratospheric injection but the upper-troposphere is also affected, and this is even before and after the paroxysmal stratospheric event. This can explain the larger aerosol extinctions at lower altitudes in the belt 25-60°S. Of course we cannot exclude the impact of residual cirrus clouds extinction at these altitudes but: 1) the SAGEIII Ångström exponent values have relatively large average values, which is more an indication of aerosols than cirrus clouds, and 2) we don't see any reason to have a specific cirrus clouds contamination only in this band and not in the 15-25°S and 60-80°S belts.

11)  p12, Figure 5: the y-axis text is too long and too small. Why not simply: Daily average RF (W/m2) ... and the rest is then explained in the figure caption.

Done. Figure 5 has been modified according to this comment and Reviewer #1's SC4.

12) Also the x-axis text could be enlarged. a), b), c)....e) in the panels are larger than the x- axis text! That is not optimum.

Figure 5 is now larger and has optimised blank spaces and aspect ratio, so to make x-axis (and y-axis) labels more readable.

13) p13, l297-l328: What do we learn? ... if we do not trust the extinction observations?

We don't understand the question. We indeed trust the extinction observations, with the limitation discussed above (SC8-9).

14) p14, Figure 6, again, February to April values show a coherent development.... but January results are very different. Underestimation of smoke extinction values?

15) p15, Table 1, again the same problem, why are the January values not in line with the rest (Feb to April data).

16) p16, Table 2, the same problem.

See reply to SC8-9.

17) Please, improve y-axis text also here (too long, too small).

Done, we changed y-axis text and general figure aspect ratio and organisation